# Myricetin Nanofibers Enhanced Water Solubility and Skin Penetration for Increasing Antioxidant and Photoprotective Activities

**DOI:** 10.3390/pharmaceutics15030906

**Published:** 2023-03-10

**Authors:** Tzu-Ching Lin, Chun-Yin Yang, Tzu-Hui Wu, Chih-Hua Tseng, Feng-Lin Yen

**Affiliations:** 1School of Pharmacy, College of Pharmacy, Kaohsiung Medical University, Kaohsiung City 807, Taiwan; 2Department of Pharmacy and Master Program, Collage of Pharmacy and Health Care, Tajen University, Pingtung County 90741, Taiwan; 3Drug Development and Value Creation Research Center, Kaohsiung Medical University, Kaohsiung City 807, Taiwan; 4Department of Fragrance and Cosmetic Science, College of Pharmacy, Kaohsiung Medical University, Kaohsiung City 807, Taiwan; 5Department of Medical Research, Kaohsiung Medical University Hospital, Kaohsiung City 807, Taiwan; 6Department of Pharmacy, Kaohsiung Municipal Ta-Tung Hospital, Kaohsiung City 801, Taiwan; 7Institute of Biomedical Sciences, National Sun Yat-Sen University, Kaohsiung City 804, Taiwan; 8College of Professional Studies, National Pingtung University of Science and Technology, Pingtung County 912, Taiwan

**Keywords:** myricetin, water solubility, skin penetration, myricetin nanofiber, antioxidant, photoprotective

## Abstract

Excessive exposure to ultraviolet radiation (UV) can induce oxidative stress through the over-production of reactive oxygen species (ROS) on the skin. Myricetin (MYR), a natural flavonoid compound, significantly inhibited UV-induced keratinocyte damage; however, its bioavailability is limited by its poor water solubility and inefficient skin penetration ability, which subsequently influences its biological activity. The purpose of the study was to develop a myricetin nanofibers (MyNF) system of hydroxypropyl-β-cyclodextrin (HPBCD)/polyvinylpyrrolidone K120 (PVP)-loaded with MYR that would enhance the water solubility and skin penetration by changing the physicochemical characteristics of MYR, including reducing the particle size, increasing the specific surface area, and amorphous transformation. The results also revealed that the MyNF can reduce cytotoxicity in HaCaT keratinocytes when compared with MYR; additionally, MyNF had better antioxidant and photoprotective activity than raw MYR for the UVB-induced HaCaT keratinocytes damage model due to the MyNF increased water solubility and permeability. In conclusion, our results demonstrate that MyNF is a safe, photostable, and thermostable topical ingredient of antioxidant nanofibers to enhance the skin penetration of MYR and prevent UVB-induced skin damage.

## 1. Introduction

The World Health Organization (WHO) estimates that approximately 1.5 million skin cancer cases were diagnosed worldwide in 2020. Overexposure to ultraviolet radiation (UV) is the main cause of a number of skin diseases and UVB rays are principally responsible for sunburns and skin cancer [1]. Previous studies have demonstrated that excessive UVB interacts with water in the skin to increase the intracellular reactive oxygen species (ROS) levels of epidermal keratinocytes, thus disrupting the cellular antioxidant defense system and resulting in skin photodamage [2]. Natural products have been used as potent cosmeceutical substances to achieve the photoprotective effect. Myricetin (MYR), a hexahydroxyflavonoid substituted by hydroxyl groups (-OH) at the 3, 3′, 4′, 5, 5′ and 7 positions (Figure 1), has been to shown to be abundantly distributed in the bark of bayberry (*Myrica rubra*), vegetables, honey, tea [3], berries, and red wine [4]. MYR has a variety of pharmacological activities, such as antioxidant, antiinflammatory, antiphotoaging, antiallergic, and antimicrobial activities [5]. However, a significant drawback of MYR is that it is poorly soluble in water (water solubility: 16.6 µg/mL) and the US Pharmacopoeia determined that MYR was a practically insoluble active pharmaceutical ingredient (API). It is well known that MYR can dissolve well in organic solvents, e.g., methanol, ethanol, and DMSO [6,7], but these organic solvents have the potential to cause skin diseases, including acute irritant dermatitis, chronic eczema, and allergic contact dermatitis [8]. On the other hand, MYR had poor skin absorption due to its physicochemical properties, such as a crystalline form, a melting point higher than 200 °C (357 °C), and skin penetration coefficient of less than −1 (Log Kp: −7.4 cm/s). Therefore, these above disadvantages could be limited to the application of MYR in the pharmaceutical and cosmetic industries.

Based on our best knowledge, drug delivery systems (DDS), such as microemulsions, nanoemulsions, polymeric nanoparticles, liposomes, nanogel, and nanofibers, are usually used to increase their absorption of API through water solubility enhancement, skin permeation increment, and physicochemical characteristics improvement [9]. Nanofibers stand out among these DDS because of their larger surface area, smaller pore size, higher porosity, lower density, and lack of solvent residue. In addition, nanofibers can be formed in a variety of ways, including electrospinning, molecular assembly, and thermally induced phase separation [10]. Electrospinning is an effective technology for generating continuous nanofibers by electrostatic force. The advantages of electrospun nanofibers are the ability to mass-produce continuous nanofibers from different polymers and produce ultrathin polymer fibers with controllable compositions, diameters, and orientations, which are usually used to load hydrophobic drugs for improving dissolution due to their much greater specific surface area per unit mass [11]. Nontoxic nanocarriers are the first option for secure nanofiber compositions [12]. Previous studies indicate that 2-hydroxypropyl-β-cyclodextrin (HPBCD) is an important cyclodextrin derivative that can be used as a solubilizer, encapsulation media, and stabilizer in a variety fields, including pharmaceuticals, food, and cosmetics for improving the poor water solubility and stability of the drug molecule [13]. Polyvinylpyrrolidone K120 (PVP) is also a nontoxic water-soluble polymer obtained by monomer N-vinylpyrrolidone that helps to encapsulate both hydrophilic and lipophilic drug as well as improve photostability and thermostability in the formulation [14]. Additionally, it has been shown that PVP could have a certain influence on the solubilization effect of HPBCD [15].

The purpose of the study was to use electrospinning technology to prepare optimal HPBCD/PVP-loaded myricetin nanofibers (MyNF) and to compare the physicochemical characteristics between MYR and MyNF for elucidating the improvement mechanism of water solubility and skin absorption. Moreover, we also compared the antioxidant capacity and cell viability of MYR and MyNF for confirming their photoprotective effect in the UVB-induced HaCaT keratinocytes damage model.

## 2. Materials and Methods

### 2.1. Materials

Myricetin (MYR) with 98% of purity was purchased from Baichuan Kanze (Shaanxi, China). 2-Hydroxypropyl-β-cyclodextrin (HPBCD) with an average molar substitution degree of 4.75 per anhydroglucose unit (average molecular weight 1410 g/mol) was purchased from Zibo Qianhui (Shandong, China). Polyvinylpyrrolidone K120 (PVP, average molecular weight 3,000,000 g/mol), 2,2-diphenyl-1-picrylhydrazyl (DPPH), 2,2′-Azino-bis-[3-ethylbenzthiazoline-6-sulfonic acid] (ABTS), potassium peroxodisulfate (K_2_S_2_O_8_), trichloroacetic acid (TCA), potassium hexacyanoferrate(III) (K_3_[Fe(CN)_6_]), L-Ascorbic acid, 2′,7′—dichlorofluorescin diacetate (DCF-DA) were purchased from Sigma-Aldrich (St. Louis, MO, USA). Sodium phosphate dibasic (Na_2_HPO_4_) and Sodium dihydrogen phosphate (NaH_2_PO_4_·H_2_O) were purchased from J.T. Baker Chemical (Warren, NJ, USA). Iron(III) chloride hexahydrate (FeCl_3_·6H_2_O) was purchased from Nihon Shiyaku (Kyoto, Japan). Potassium dihydrogen phosphate (KH_2_PO_4_) and dipotassium phosphate (K_2_HPO_4_) were purchased from Ferak (Berlin, Germany). HaCaT human keratinocytes cell line was obtained from the Istituto Zooprofilattico Sperimentale della Lombardia edell’Emilia Romagna (Brescia, Italy). Dulbecco’s modified Eagle’s medium (DMEM) was purchased from Himedia Laboratories (Mumbai, India). Fetal bovine serum (FBS) was obtained from Thermo Fisher Scientific (Waltham, MA, USA). Penicillin, streptomycin, and amphotericin B solution (PSA) were purchased from Biological Industries (Connecticut, NE, USA). 3-(4,5-cimethylthiazol-2-yl)-2,5-diphenyl tetrazolium bromide (MTT) was purchased from MDBio (Taipei, Taiwan).

### 2.2. HPLC Analysis of MYR

The calibration curve of MYR was determined by using high-performance liquid chromatography (HPLC). The HPLC instrument (LaChrom Elite L-2000, Hitachi, Tokyo, Japan) equipped with a L-2130 pump, a L-2200 autosampler, and a L-2420 (UV–vis) detector connecting with a Mightysil RP-18 GP column (250 × 4.6 mm, 5 µm; Kanto Chemical Co., Inc., Tokyo, Japan). The mobile phase was composed of methanol and 0.1% phosphoric acid solution in a fixed ratio (55:45; *v*/*v*). The system was set at 1 mL/min for flow rate, the injection volume was 10 μL and the wavelength was detected at 360 nm. MYR was dissolved in methanol and then diluted to nine concentrations (0.005–100 µg/mL) as the standard in order to determine the calibration curve. The retention time peak of MYR was observed at 5.40 min. The MYR calibration curve is used to determine the MYR content of each MyNF.

### 2.3. Preparation of HPBCD/PVP-Loaded Myricetin Nanofibers (MyNF)

MyNF with different ratios of MYR:HPBCD:PVP (1:20:4, 1:20:8, and 1:20:12; *w*/*w*/*w*) were prepared using a modified version of the method described in [12]. Firstly, MYR, HPBCD, and PVP were added into 10 mL ethanol and then stirred with a magnetic stirrer for 90 min. Subsequently, the clear solution was transferred to a 10 mL syringe with a blunt end needle (22 gauge, 0.42 mm diameter) and then placed on an auto-injection micropump. The injection rate was set at 0.02 mL/min and the electrospinning process was operated under an applied voltage of 18 kV. An aluminum foil was placed on the collector at a distance of 11 cm from the needle tip. The freshly synthesized nanofibers with aluminum foil were placed in a sealed plastic bag and stored in a moisture-proof box before analysis.

### 2.4. Water Solubility

Pure MYR (1 mg) and different ratio formulations of MyNF (containing 1 mg MYR) were dissolved in 1 mL deionized water and shaken (Vortex-Gene 2, Scientific Industries, Bohemia, NY, USA) for 10 min, respectively. Subsequently, each sample was filtered with a 0.45 µm syringe filter (13 mm Acrodisc^®^ syringe filters with GHP membrane, Pall Corporation, Port Washington, NY, USA) and then diluted 100-fold with pure water before HPLC analysis, after which the MYR concentration of each sample was calculated by the MYR standard curve.

### 2.5. Yield and Encapsulation Efficiency

Firstly, different ratios MyNF (containing 1 mg MYR) were weighed and placed in a 1.5 mL tube with 1 mL methanol and shaken for 10 min, respectively. The MYR concentration in each sample was evaluated using the HPLC analysis method mentioned above. The encapsulation efficiency of different ratio MyNF formulations (containing 1 mg MYR) was weighed and dissolved with 1 mL distilled water. After that, 500 µL of each sample was placed in a high-speed centrifugal filter device (Nanosep^®^ Centrifugal Devices with Omega™ Membrane molecular weight 10,000, Pall Corporation, Port Washington, NY, USA) and centrifuged at 10,000 rpm (Centrifuge 5430 R; Eppendorf, Hamburg, Germany) for 10 min. The MYR concentration in each sample was evaluated using the HPLC analysis method mentioned above. The yield and encapsulation efficiency of MyNF were calculated using the following equations (Equations (1) and (2), respectively):(1)$$\mathrm{Yield}\;(\%)=\frac{\mathrm{actual}\;\mathrm{amount}\;\mathrm{of}\;\mathrm{MYR}\;\left(\mathrm{mg}\right)}{\mathrm{theoretical}\;\mathrm{amount}\;\mathrm{of}\;\mathrm{MYR}\;\left(\mathrm{mg}\right)}\;\times \;100\%\;$$
(2)$$\mathrm{Encapsulation}\;\mathrm{efficiency}\;(\%)=\frac{\mathrm{theoretical}\;\mathrm{amount}\;\mathrm{of}\;\mathrm{MYR}\;\left(\mathrm{mg}\right)-\mathrm{free}\;\mathrm{amount}\;\mathrm{of}\;\mathrm{MYR}\;\left(\mathrm{mg}\right)}{\mathrm{theoretical}\;\mathrm{amount}\;\mathrm{of}\;\mathrm{MYR}\;\left(\mathrm{mg}\right)}\;\times \;100\%$$

### 2.6. Determination of Particle Size and Morphology by SEM, DLS and TEM

The surface morphology of MYR, excipients, and MyNF were observed by scanning electron microscopy (SEM; Hitachi S4700, Hitachi, Tokyo, Japan). Each sample was sputter-coated with a thin gold layer using an ion sputter coater (Hitachi E-1045, Hitachi, Tokyo, Japan) before SEM analysis. The particle size of MYR and each MyNF was measured by a dynamic light scattering particle size analyzer (DLS; ELSZ-2000; Otsuka Electronics, Osaka, Japan). The sample preparation for particle size measurement involved dissolving each formulation (containing 1 mg MYR) in 1 mL distilled water and then diluted it with 10-fold pure water and immediately placed in a cuvette for determining the particle size. Additionally, the particle morphology of MyNF was observed using a transmission electron microscope (TEM; JEOL JEM-2000 EXII; JEOL, Tokyo, Japan). MyNF (containing 1 mg MYR) was dissolved in 1 mL distilled water and diluted 100-fold with distilled water, and then dropped into the carbon-coated copper grid and stained with 0.5% (*w*/*v*) phosphotungstic acid (Sigma, St. Louis, MO, USA). Each sample was kept in a moisture-proof container until TEM observation.

### 2.7. Determination of Crystalline-to-Amorphous Transformation by XRD

X-ray diffractometry (XRD; Siemens, Munich, Germany) was used to determine the crystalline and amorphous structure of MYR and MyNF with Cu Kα radiation at 40 kV and a current of 40 mA. The incident and the diffracted beam were at 2θ and scanned ranges were from 2° to 50°.

### 2.8. Determination of Hydrogen Bond Formation by FTIR and ^1^H-NMR

Fourier transform infrared spectroscopy (FTIR; Alpha II, Bruker, Billerica, MA, USA) and 600 MHz ^1^H nuclear magnetic resonance (^1^H-NMR; Oxford Instrument, Abingdon, UK) were used to identify functional group interaction and hydrogen-bond formation between MYR and excipients. A total of 9 mg of potassium bromide (Sigma, St. Louis, MO, USA) was mixed with 1 mg of MYR, and different ratios of MyNF, HPBCD, and PVP, respectively. Then, each sample was compressed as a thin tablet and placed into the FTIR analyzer. The FTIR scanning range was set from 400 to 4000 cm^−1^ for identifying the chemical functional groups. For the ^1^H-NMR spectra, 5 mg of each sample was first dissolved in 0.5 mL of dimethyl sulfoxide-*d*_6_ (DMSO-*d*_6_; Sigma-Aldrich, St. Louis, MO, USA), and then placed into the magnetic field. NMR spectroscopy is produced by the excitation of nuclei with electromagnetic radiation.

### 2.9. Stability of MYR and MyNF

The sample preparation was performed of pure 1 mg of MYR dissolved in 1 mL of methanol and MyNF (MYR:HPBCD:PVP at a 1:20:12 ratio containing 1 mg of MYR) dissolved in 1 mL deionized water before stability assay. The photostability test was performed using the approach of Huang et al. with suitable modifications [16]. Briefly, each sample was exposed to UV radiation of 560 µW/cm^2^ for 0, 3, 6, and 9 days with the corresponding energies of 0, 145, 290, and 435 J/cm^2^, respectively. The thermostability and centrifugation test was performed as described with some modifications [17]. All samples were passed through the cycles of accelerating temperature tests at −20 °C, 4 °C, 25 °C, 50 °C, and 65 °C. Samples were placed in each temperature condition for 24 h. For the centrifugation test, each sample was respectively centrifuged at 4000 and 10,000 rpm for 10 min using Centrifuge 5430 R (Eppendorf, Hamburg, Germany). After that, the supernatant of each sample was diluted 100-fold with pure water and filtered with a 0.45 µm syringe filter before HPLC analysis, after which the MYR content in each sample was calculated.

### 2.10. In Vitro Skin Penetration Assay

In vitro skin penetration assay was carried out using a modified methodology from the European Cosmetic and Perfumery Association (COLIPA) [18]. Pork skin was obtained from a local market and immediately cut into an area of 4 cm^2^ (2 cm × 2 cm), and the available diffusion area was 0.785 cm^2^. The skin permeation of MYR through pork skin was determined using Franz diffusion cell (FDC). The pork skin was placed between the donor (upper) and receptor (lower) chambers of the FDC system. Firstly, the receptor chambers were filled with buffer solution (0.14 M NaCl, 2 mM K_2_HPO_4_, 0.4 mM KH_2_PO_4_, and pH 7.4) and then placed a magnetic stirring bar at the bottom of the receptor chamber. The FDC system was maintained at 32 °C under the circulating water system and stirred at 600 rpm. Next, 200 µL of 1 mg/mL of MYR and MyNF (MYR:HPBCD:PVP at a 1:20:12 ratio containing 1 mg of MYR) were loaded on the pork skin of donor chambers, respectively. The sampling times for topical administration were determined at 1, 2, 4, and 8 h. After that, the stratum corneum of the skin sample was obtained by 15-time tape stripping with 3 M Transpore tape. The epidermal and dermal layers were separated using a scalpel and sliced into tiny pieces. Each sample was then put into a methanol-filled tube, where the MYR was extracted using a sonicator for 1 h. The content of MYR in each sample was analyzed by the HPLC method mentioned above.

### 2.11. DPPH and ABTS^+^ Free Radical Scavenging Activity

The DPPH and ABTS^+^ free radical scavenging activity were evaluated antioxidant capacities of the sample according to Lin et al. [13]. The experimental groups were pure MYR in water, pure MYR in methanol, MyNF in water, and vitamin C in water (as a positive antioxidant). Briefly, 100 µL of 200 µM DPPH free radical solution and 100 µL of each sample were mixed in each well of the 96-well plate for 30 min. The absorbance was measured using a microplate spectrophotometer at 517 nm. For the ABTS assay, the ABTS^+^ free radical solution was first generated with 7 mM ABTS in water and 2.45 mM potassium persulfate in water at a 2:1 ratio (*v*/*v*) and maintained in a fridge away from light for 12–16 h until the reaction completed. Then, 170 µL of ABTS^+^ free radical solution and 30 µL of each sample were mixed in each well of the 96-well plate for 30 min. The absorbance was measured using a microplate spectrophotometer at 734 nm. The DPPH and ABTS^+^ radical scavenging activity was calculated using the following equations (Equations (3) and (4), respectively):(3)$$\mathrm{DPPH}\;\mathrm{free}\;\mathrm{radical}\;\mathrm{scavenging}\;\mathrm{activity}\;(\%)=\frac{\mathrm{OD}517\;\mathrm{of}\;\mathrm{control}-\mathrm{OD}517\;\mathrm{of}\;\mathrm{sample}\;}{\mathrm{OD}517\;\mathrm{of}\;\mathrm{control}}\;\times \;100\%$$
(4)$${\mathrm{ABTS}}^{+}\;\mathrm{free}\;\mathrm{radical}\;\mathrm{scavenging}\;\mathrm{activity}\;(\%)=\frac{\mathrm{OD}734\;\mathrm{of}\;\mathrm{control}-\mathrm{OD}734\;\mathrm{of}\;\mathrm{sample}\;}{\mathrm{OD}734\;\mathrm{of}\;\mathrm{control}}\;\times \;100\%\;$$

### 2.12. Ferric Reducing Antioxidant Power

Briefly, 50 µL of each sample, 50 µL of phosphate buffer (0.2 M, pH 6.6), and 50 µL of K_3_Fe(CN)_6_ (1%) were mixed in the 1.5 mL microtube and heated to 50 °C for 20 min, followed by a cool water bath for 2 min. The mixtures were added with 50 µL of trichloroacetic acid (10%) before centrifuging for 10 min at 3000 rpm and 4 °C. Next, 100 µL of each supernatant was transferred into each well of a 96-well plate and then 100 µL of deionized water and 10 µL of FeCl_3_·6H_2_O (0.1%) were added and mixed for 10 min. Finally, the absorbance of each sample was determined by a microplate spectrophotometer at 700 nm.
(5)$$\mathrm{Reduction}\;\mathrm{power}=\frac{\mathrm{OD}700\;\mathrm{of}\;\mathrm{sample}-\mathrm{OD}700\;\mathrm{of}\;\mathrm{control}\;}{\mathrm{OD}700\;\mathrm{of}\;\mathrm{control}}\;\times \;100\%\;$$

### 2.13. Cell Culture and Cell Safety Assay

The HaCaT keratinocytes were maintained in DMEM medium (containing 10% FBS and 1% PSA) and cultured in an incubator under 37 °C with 5% of CO_2_ (Thermo Fisher Scientific, Waltham, MA, USA). For cell safety assay of MYR and MyNF, 100 µL of medium containing 1 × 10^4^ of HaCaT cells were seeded into each well of 96 well plate for 24 h. Next, the culture medium of each well was removed and then added different concentrations (20 to 320 μM) of MYR in PBS, MYR in DMSO, MyNF (MYR:HPBCD:PVP, 1:20:12) in PBS, and blank (HPBCD:PVP; 20:12; *w*/*w*) in PBS with serum-free DMEM medium for 24 h, respectively. After 24 h, the culture medium was removed and washed with 100 μL of PBS. In total, 150 μL of MTT (0.5%) solution was added into each well and placed in an incubator for 3 h. After that, MTT solution of each well was removed, and then 100 μL of DMSO was added to dissolve the purple formazan crystals. The absorbance was measured using a microplate spectrophotometer at 550 nm.
(6)$$\mathrm{Cell}\;\mathrm{viability}=\frac{\mathrm{OD}550\;\mathrm{of}\;\mathrm{sample}\;}{\mathrm{OD}550\;\mathrm{of}\;\mathrm{control}}\;\times \;100\%$$

### 2.14. Photoprotective Effects of MYR and MyNF on HaCaT Cells

The photoprotective effects of MYR and MyNF against UVB-induced damage on HaCaT keratinocytes were determined according to Yang et al. [12]. Firstly, 100 µL of medium containing 1.5 × 10^4^ HaCaT cells were seeded in each well of a 96 well plate for 24 h. Next, the medium was removed and immediately washed with 100 µL of PBS. Then, 20 to 160 μM of each sample with serum-free DMEM medium were added into each well, respectively. After 3 h of incubation, the medium of each well was removed and then 100 μL of PBS was added before irradiation of UVB (240 mJ/cm^2^). After that, PBS was removed and added to different concentrations (20 to 160 μM) of the test sample for 24 h. The cell viability of each sample was determined by the MTT method mentioned above.

### 2.15. Effects of MYR and MyNF on UVB-Induced Reactive Oxygen Species (ROS) Assay

The ROS level was measured by DCFH-DA assays. Firstly, 100 µL of medium containing 1 × 10^4^ HaCaT cells were seeded in each well of the black 96 well plate. After 24 h of incubation, the medium was removed and washed with 100 µL of PBS before being pretreated with 100 μL of test samples for 24 h. Then, the test sample in each well was removed, washed with 100 μL of PBS, and 100 μL of 10 μM DCFH-DA was added before being irradiated with UVB (240 mJ/cm^2^). After 30 min, the DCFH-DA solution was removed, washed twice with PBS, and soaked in PBS. Finally, the ROS production was measured using excitation and emission wavelengths of 458/528 nm by the fluorescent plate reader (BioTek, Winooski, VT, USA).

### 2.16. Statistical Analysis

Experimental data were presented as mean ± standard deviation using Microsoft Excel 2019 software (Microsoft Office; Microsoft Corporation, Redmond, WA, USA). The statistical significance of experimental group differences was determined by a follow-up one-way analysis of variance (ANOVA) by running Tukey’s post-hoc test, and *p*-value ≤ 0.05 are conventionally regarded as statistically significant.

## 3. Results

### 3.1. Water Solubility, Yield, and Encapsulation Efficiency of MYR and Its Nanofibers Formulation

The water solubility, yield, and encapsulation of MyNF formulations at different ratios are shown in Table 1. The water solubility of MYR was 0.32 µg/mL, which means it is a practically insoluble compound (water solubility: <0.1 mg/mL) based on the water solubility classification of U.S. Pharmacopoeia [6]. Table 1 also indicated that the water solubility of MYR was effectively raised by 2665, 2787, and 2858 times, respectively, by MYR:HPBCD:PVP formulations at 1:20:4, 1:20:8, and 1:20:12 (*w*/*w*/*w*). These findings suggest that the MYR/HPBCD/PVP nanofibers formulations are critical for improving water solubility of MYR, and then the ratio at 1:20:12 has the most potential for enhancing the water solubility. Moreover, the yield of nanofibers showed a positive correlation with water solubility and an obvious proportional dependence on the PVP content. The encapsulation efficiency of all three MYR nanofibers formulations is above 98%. The above results indicate that MYR can be effectively encapsulated into a nanofibers system using HPBCD with PVP co-carriers. Particularly, MyNF at a ratio of 1:20:12 might be an optimal formulation with which to improve the poor water solubility of MYR.

### 3.2. Morphology and Size of MYR and MyNF

#### 3.2.1. Surface Morphology of MYR and MyNF by SEM

In our research, we used scanning electron microscope (SEM) imaging to visualize the morphology of MYR, excipients, and MyMF to elucidate their surface characteristics (Figure 2). It can be seen that the appearance of pure MYR is a rectangular crystalline powder with a length of about 10–40 μm and a width of about 2–4 μm (Figure 2A). The pure HPBCD had a smooth sphere and porous structure (Figure 2B). The pure PVP had plate-like structures (Figure 2C). The MyNF transformed into smooth fibers and had a high specific surface area (Figure 2D–F). The irregular shape of MYR, as well as the original HPBCD and PVP, were invisible. These results suggest that MYR nanofibers have been synthesized and that MYR was merged into HPBCD with PVP. In addition, the fiber diameters of MyNF with different ratios were 276.59 ± 16.36, 561.47 ± 67.09, and 621.87 ± 35.33 nm, respectively, for MYR:HPBCD:PVP formulations at 1:20:4, 1:20:8, and 1:20:12 (*w*/*w*/*w*). It can be found that the fiber diameters of nanofibers showed a positive correlation with a proportional dependence on the PVP content. When MyNF had the ratios of MYR:HPBCD:PVP at 1:20:4 and 1:20:8 ratios (*w*/*w*/*w*), the fiber shape led to significant nodules or beads along its length (Figure 2D,E), and their average diameters were lower than 1:20:12. As mentioned above, it was found that a higher ratio of PVP (1:20:12) led to MyNF having a larger fiber, and as the PVP ratio increased, fibers were shown to exhibit a continuous, smooth, and bead-free fiber surface (Figure 2F).

#### 3.2.2. Particle Size and Polydispersity Index of MYR and MyNF by DLS

We used dynamic light scattering (DLS) to analyze the particle size and polydispersity index (PDI) of MYR and different ratios of MYR:HPBCD:PVP (1:20:4, 1:20:8, and 1:20:12; *w*/*w*/*w*) reconstituted in water to elucidate their particle size or diameter and distribution or aggregation. It is well known that the polydispersity index (PDI) is usually used to define the particle size distribution. A polydispersity index value closer to 0.01 indicates a monodisperse particle size distribution, whereas a PDI closer to 1.0 indicates a formulation with a broad particle size distribution, and when the PDI is greater than 1.0, the formulation presents multiple particle distributions. For polymer-based nanomaterials, PDI values of 0.3 and below are most frequently regarded as acceptable in practice [19]. As shown in Table 2, the average particle size diameter and polydispersity index (PDI) of MYR were, respectively, 4065.27 ± 327.11 nm and 1.37 ± 0.10, which indicated that the particle size of MYR had multiple distributions. In contrast, the particle size of MyNF formulations at 1:20:4, 1:20:8, and 1:20:12 ratios (*w*/*w*/*w*) were 137.33 ± 6.79, 88.73 ± 3.07, and 38.27 ± 1.02 nm, respectively; it can be seen the particle size of MyNF decreased in a PVP-dependent manner. The PDI of MyNF formulations at 1:20:4, 1:20:8, and 1:20:12 ratios (*w*/*w*/*w*) were 1.37 ± 0.01, 0.07 ± 0.00, and 0.24 ± 0.02 nm, respectively; the PDI value of all three was below 0.3, which is considered to be within an acceptable range for uniform particle dispersion.

#### 3.2.3. Particles Morphology of MyNF under TEM Observation

The particle morphology from transmission electron microscope clearly observed round colloidal droplets of MYR/HPBCD/PVP with a particle size of 400–2000 nm (Figure 3A). Figure 3B shows that many MYR/HPBCD inclusion particles are dispersed in the PVP colloidal system and present round droplets. In Figure 3C, MYR/HPBCD inclusion particles below 10 nm can be seen more clearly.

### 3.3. Crystalline and Amorphous Structure of MYR and MyNF

X-ray diffractometry (XRD) is widely applied in pharmaceutical research, which can be used for the characterization of the composition and crystalline form of a drug delivery system [20]. The XRD patterns of MYR, HPBCD, PVP, their physical mixture, and different ratios of MyNF are shown in Figure 4. The XRD pattern of MYR displayed several characteristic diffraction peaks at 5.64°, 7.63°, 9.23°, 11.25°, 13.95°, 16.46°, 23.32°, 23.58°, 28.24°, and 29.41°. A broad peak centered at 18.14° and 10.75° was observed in HPBCD and PVP, respectively. The physical mixture of MYR, HPBCD, and PVP did not interact chemically as well as retaining their original physical characteristics. For the XRD pattern of MyNF, the diffraction peaks of MYR disappeared and exhibited a similar pattern to HPBCD and PVP, indicating that the crystalline form of MYR had been converted to an amorphous form as well as showing MYR might be entirely incorporated with HPBCD and PVP.

### 3.4. Intermolecular Chemical Bond Formation between MYR and Its Excipients 

#### 3.4.1. Fourier-Transform Infrared Spectroscopy

Fourier-Transform Infrared (FTIR) spectroscopy is beneficial for investigating the intermolecular bonds between of active ingredients and excipients. The FTIR spectrum of MYR, HPBCD, PVP, their physical mixture, and different ratios of MyNF are shown in Figure 5. The FTIR spectra of MYR displayed several characteristic stretching vibrations for C-C stretching of A ring at 1660 cm^−1^, C=O stretching of C ring at 1607 cm^−1^, C=C stretching of A ring at 1521 cm^−1^, C-C stretching at 1381 cm^−1^, C-C stretching of B ring at 1330 cm^−1^, and C-O stretching of B ring at 1028 cm^−1^. The HPBCD detected functional groups were O-H stretching at 3388 cm^−1^, C-H stretching at 2927 cm^−1^, C-H stretching at 2927 cm^−1^, H-O-H bending at 2927 cm^−1^, C-O stretching at 1156 cm^−1^, and C-O-C stretching at 1034 cm^−1^, which were almost comparable to the previous study of by Han et al. [21]. The PVP showed O-H stretching at 3454 cm^−1^, C-H stretching at 2957 cm^−1^, C=O stretching at 1662 cm^−1^, C-H stretching at 1434 cm^−1^, and C-N stretching at 1287 cm^−1^, which showed a similar functional group as Yang et al. [12]. The physical mixture of MYR, HPBCD, and PVP almost characteristic stretching vibration were obvious, which demonstrated that there was essentially no interaction among MYR, HPBCD, and PVP when the three chemicals were mixed physically. In FTIR spectra of MyNF, the original absorption bands of MYR at 1660, 1607, and 1521 cm^−1^ shifted to a broad band at 1600~1700 cm^−1^, demonstrating that the aromatic A ring of O-H and C ring of C=O groups of MYR probably interacted with HPBCD and PVP. Additionally, the O-H stretching of MyNF was broadened at 3200~3500 cm^−1^, indicating MYR was successfully encapsulated into HPBCD with PVP through the formed hydrogen bond interaction.

#### 3.4.2. ^1^H-Nuclear Magnetic Resonance Spectroscopy

The intermolecular hydrogen bond formation between MYR, blank (HPBCD:PVP; 20:12; *w*/*w*), and MyNF (MYR:HPBCD:PVP; 1:20:12; *w*/*w*/*w*) was determined by ^1^H-Nuclear Magnetic Resonance (^1^H-NMR) spectroscopy. In Figure 6A, the ^1^H-NMR spectrum of raw MYR showed aromatic at *δ* 6.174 (1H, d, *J* = 2.0 Hz, H-6), 6.366 (1H, d, *J* = 2.0 Hz, H-8), 7.232 (2H, s, H-2′, 6′), hydroxyl proton at 12.480 (1H, s, OH-6), and another lumped together of five hydroxyl protons at 9.279 (5H, brs, OH-3, 3′, 4′, 5′, 7) ppm, indicating the similar functional group to results of Yao et al. [22]. In Figure 6B, the resonance peaks for blank (HPBCD:PVP; 20:12; *w*/*w*) were found in the regions of *δ* 1.0~6.0 ppm. In Figure 6C, the ^1^H-NMR spectrum of MyNF showed that the protons of MYR shifted at *δ* 6.175 (1H, d, *J* = 1.8 Hz, H-6), 6.374 (1H, d, *J* = 1.8 Hz, H-8), 7.238 (2H, s, H-2′, 6′), 12.464 (1H, s, OH-6), and another hydroxyl signal at 9.219 (5H, brs, OH-3, 3′, 4′, 5′, 7) ppm. As shown in Figure 6D, the chemical shifts variation (Δ*δ*_H_ in ppm) of H-6, H-8, H-2′, 6′, and OH-5 values are ≤ −0.016 ppm, which indicated the MYR with HPBCD/PVP inclusion complex had a negligible difference. Then, Δ*δ*_H_ of OH-3, 3′, 4′, 5′, and 7 are −0.06 ppm, indicating that the hydroxyl protons had shifted. In addition, the integral value (*∫*_H_) of the hyroxyl proton (OH-3, 3′, 4′, 5′, and 7) signal of MYR and MyNF (MYR in the MYR/HPBCD/PVP inclusion complex) are, respectively, 4.61 and 1.74, and the integral value variation (Δ*∫*_H_) of the hyroxyl proton (OH-3, 3′, 4′, 5′, and 7) is 2.87. These results were similar to reported by Yang et al. [23], the entire B ring and part of the C ring of myricetin were included in the HPBCD cavity through hydrogen bond interaction. Based on previous studies, we deduced a probable inclusion mode for the MYR/HPBCD/PVP inclusion complex, as protons on aromatic B and C rings of MYR effectively formed an intermolecular hydrogen bond with HPBCD/PVP excipients.

### 3.5. Photostability, Thermostability, and Centrifuge Testing of MYR and MyNF

It is well known that product stability, such as photostability, thermostability, and shipping process, impacts biological activity, product safety, and quality. The photostability of MYR and MyNF is shown in Figure 7A, MYR and MyNF were, respectively, dissolved in DMSO and water and started degradation after 3 day of light exposure in a time-dependent manner. We also evaluated MYR and MyNF powder after 9 days of light exposure. The MYR content of pure MYR powder and MyNF retained 71.60 ± 2.18% and 94.32 ± 1.94%, respectively. These results indicated that the degradation of MyNF powder was lower than the MYR power, MYR solution, and MyNF solution; therefore, MyNF had the best photostability in dry form, but not in solution form.

#### 3.5.1. Thermostability Testing

The thermostability of MYR and MyNF was determined by the accelerated temperature cycling for evaluating the degradation of myricetin in each sample. As shown in Figure 7B, the degradation rate of MYR solution, MyNF solution, MYR powder, and MyNF were 76.34 ± 3.50%, 90.36 ± 1.76%, 93.88 ± 1.16%, and 98.39 ± 3.59%, respectively. These results indicate that MyNF exhibits excellent thermostability in dry powder form to prevent the thermal degradation of MYR.

#### 3.5.2. Centrifuge Testing

Moreover, centrifuge testing is usually used to evaluate whether shipping movements may cause active substance degradation or formulation separation. As shown in Figure 7C, there was no significant difference between the MYR solution and MyNF solution at 4000 rpm. After centrifugation at 10,000 rpm, the degradation rate of MYR solution and MyNF solution were 84.03 ± 5.10% and 99.93 ± 0.61%, respectively. These results indicate that MyNF exhibits excellent stability of centrifugation in solution form to prevent forced degradation of MYR.

### 3.6. In Vitro Percutaneous Penetration of MYR and MyNF

In vitro percutaneous absorption is commonly used to evaluate the deposition of active ingredients in which skin layers, including stratum corneum, epidermis, dermis, and/or systemic circulation. The results of the in vitro skin penetration of MYR and MyNF are presented in Figure 8. MYR content did not deposit in the epidermis and dermis after topical administration of pure MYR water suspension at 1, 2, 4, and 8 h. In contrast, topical administration of MyNF water solution can easily penetrate into the epidermis and dermis in a time-dependent manner, but is not found in receptor fluid. Figure 8D showed the flux profiles of MyNF and which the steady-state time at 4 h after topical administration. The result revealed that MyNF water solution is easier to penetrate skin and deposit in the epidermis and dermis when compare to pure MYR, and MyNF did not present systemic circulation after topical administration. 

### 3.7. Antioxidant Activity of MYR and MyNF

After the active ingredient is made into a drug delivery system, its biological activity may be affected; therefore, it is necessary to re-evaluate the biological activity of new active ingredient DDS. The antioxidant activities of MYR and MyNF are displayed in Table 3. In the DPPH assay, MyNF in water had the lowest SC_50_ value of DPPH free radical scavenger when compared to MYR in methanol, and vitamin C in water (*p* < 0.05). In the ABTS^+^ assay, MyNF in water not only had a similar SC_50_ value of ABTS^+^ free radical scavenger to MYR in methanol, but this value was also lower than vitamin C (*p* < 0.05). For the ferric-reducing antioxidant power (FRAP) assay, the FRAP EC_50_ value of MyNF in water was significantly lower than MYR in methanol and vitamin C in water (*p* < 0.05). On the contrary, pure MYR in water showed no effect in the three antioxidant activity assays due to its poor water solubility. According to these results, MyNF not only maintained the antioxidant activities of MYR, but also increased the antioxidant activity of MYR by improving the water solubility of MYR. 

### 3.8. Photoprotective Effect of MYR and MyNF

#### 3.8.1. Cell Survival of MYR and MyNF in UVB-Induced Keratinocytes Damage

The appropriate dermatological formulation should be low in toxicity to cells or skin. The method of cell viability assay for evaluating the safety of active substances and formulations. In addition, we also used the UVB-induced keratinocytes damage model to evaluate the photoprotective effect of MYR and MyNF at safe concentrations. As shown in Figure 9A, HPBCD and PVP are considered safe excipients, since no significant cytotoxicity was observed at all concentrations. MYR in PBS did not have any significant effect on cell viability due to the poor water solubility of MYR. MyNF in PBS at 160 μM and 320 μM displayed over 90% cell survival; however, MYR in DMSO at 160 μM and 320 μM caused 25% and 64% death of HaCaT cells at 24 h, respectively. These results indicated that MyNF in PBS is safer than raw MYR in DMSO (*p* < 0.05) and could be a safe DDS. In addition, Figure 9B showed that the photoprotective activity of raw MYR and MyNF. HaCaT keratinocytes exposure to UVB at 240 mJ/cm^2^ significantly induced over 30% cell death and presented obvious photocytotoxicity. Pretreatment with MYR in DMSO and MyNF in PBS at 10 to 80 μM significantly increased 12% to 20% cell survival in the UVB-induced phototoxicity model, but MyNF in PBS at 160 μM showed a better photoprotective effect than MYR in DMSO (*p* < 0.05). These results revealed that the photoprotective activity of MyNF was retained after the electrospinning process. Therefore, the lower concentrations of MYR and MyNF at 10 and 40 μM would be used in further ROS production assay.

#### 3.8.2. ROS Production of MYR and MyNF under UVB Irradiation

UVB overexposure can generate lots of ROS to induce oxidative stress and then cause cell death. Figure 9C revealed that the UVB irradiation group significantly increased the level of intracellular ROS production when compared to the control group (*p* < 0.05). Pretreatment with MYR in PBS cannot reduce UVB-induced ROS overproduction. Moreover, 10 and 40 μM of MYR in DMSO or MyNF in PBS can decrease ROS production after UVB irradiation. Especially, 10 μM of MyNF in PBS significantly decreased UVB-induced ROS overproduction compared with MyNF in DMSO (*p* < 0.05). These results demonstrated that MyNF at a lower concentration, such as containing 10 μM of MYR, may be more effective against photodamage by reducing ROS overproduction.

## 4. Discussion

The low skin bioavailability of myricetin due to its poor water solubility and bad skin penetration is the biggest obstacle to its application, which is also a problem faced by most other flavonoids. Lots of publications have revealed that DDS can improve the physicochemical properties of flavonoids, such as quercetin [24], baicalein [25], and naringenin [26], to increase their water solubility and skin penetration. In the current study, the myricetin nanofiber formulation (MyNF) containing MYR, HPBCD, and PVP was successfully synthesized by the electrospinning process. Table 1 indicated that the electrospinning process achieved good encapsulation efficiency, and the water solubility of MyNF was greatly improved compared with the raw MYR. The results from FTIR and ^1^H-NMR supported that HPBCD and PVP can interact with the carbonyl (-C=O) and phenolic (-OH) functional group of raw MYR to form a hydrogen-bonded intermolecular complex and resulting in improving the water solubility of MYR.

Our results also demonstrated that MyNF effectively increased the water solubility of MYR by improving the physicochemical properties. According to the result from SEM, TEM, and particle size analysis, MyNF not only reduced the particle size of MYR with a PVP ratio-dependent manner, but also increase the surface area of MYR for enhancing the water solubility of raw MYR. The ratio of 1:20:12 between MYR/HPBCD/PVP complexes was the optimal ratio of MyNF. A possible explanation for this result is that HPBCD/PVP can be used as a hydrophilic polymer carrier for solubilizing and dispersing poor water-soluble active ingredients, such as clotrimazole [27], meloxicam [28], and acyclovir [29]. In addition, the high free energy state and disordered structure of amorphous materials can significantly increase in drug solubility and dissolution rate [30]. The XRD spectra of MyNF showed an evident crystalline form of MYR that was changed to amorphous and validated the improved water solubility of raw MYR (Figure 4). For myricetin, HPBCD and PVP are a crystallization inhibitor that can increase the amorphous transformation of MYR by forming disordered structures with higher free energy, thereby improving the water solubility of MYR.

The stratum corneum (SC), a lipid matrix consisting of ceramides, fatty acids, cholesterol, cholesterol sulphate, and sterol esters in dead corneocytes, is the outset layers of our skin. The physiological function of the stratum corneum not only blocks foreign pathogens, but also affects the absorption of active ingredients [31]. Our results indicated that raw MYR cannot cross into the epidermis and dermis due to its poor water solubility and bad skin permeation. In contrast, MYR/HPBCD/PVP nanofibers can easily cross the stratum corneum and then deposit in epidermis and dermis. A possible explanation for this result is that PVP, a hydrophilic high molecular polymer, has a film-forming effect to achieve an occlusive effect on the skin, thereby increasing skin hydration of MYR and enhancing skin penetration [32]. HPBCD is not only displayed film-forming agent but can also help to deliver MYR to the skin barrier surface and partitioned from HPBCD cavity into the epidermis and dermis [16,32,33,34]. According to these reasons, MyNF dissolved as a nanoparticle film facilitates MYR deposition in the epidermis and dermis could be attributed to the film-forming, solubilization, hydration, and partitioning effects; therefore, our results suggested that the MYR/HPBCD/PVP complex is a promising approach for the MYR topical drug delivery system.

The instability of active ingredients can cause the aggregation and degradation of ingredients, thereby reducing their pharmacological activity, product safety, and quality. Stability tests, such as photostability, thermostability, and transportation stability, are usually used to evaluate the degradation and quality of active ingredients for estimating the retest period or expiration date and ensuring the effectiveness of active ingredients and the safety of the product during use. The results from stability tests showed that HPBCD/PVP can encapsulate MYR against the degradation of MYR by light, temperature, and shipping process. The previous study also indicated that HPBCD can effectively form an inclusion complex with active ingredients, such as vitamin E [35] and all-trans-retinoic acid [36], to present better stability when compared to raw ingredients.

It is very important to confirm the pharmacological activity of active ingredients after the pharmaceutical manufacturing process. In this study, several antioxidant assays, such as DPPH, ABTS, and FRAP, are used to compare the antioxidant activity of MYR and its nanofibers. The results from DPPH and FRAP assay showed the antioxidant activity of MyNF was significantly greater than raw MYR. In addition, the cell survival study mentioned that MyNF in PBS is safer than raw MYR in DMSO at higher concentrations of 160 and 320 μM (Table 3). The results showed that MyNF can reduce the cytotoxicity of MYR because MYR forms an inclusion complex with HPBCD/PVP to release MYR slowly. In the UVB-induced HaCaT keratinocytes damage model, MyNF displayed photoprotective activity by decreasing cell death and reducing ROS overproduction. Therefore, MyNF could be a potential antioxidant and photoprotective DDS for preventing skin damage after UVB overexposure.

## 5. Conclusions

The present study successfully synthesized myricetin nanofibers (MyNF), which could be effectively used to enhance the skin penetration of MYR by changing the physicochemical characteristics, including reducing the particle size, increasing the specific surface area, and amorphous transformation. The results also revealed that MyNF had lower cytotoxicity in HaCaT keratinocytes when compared to raw MYR. In addition, MyNF can promote percutaneous absorption because of nanoscale particle formation, causing increased antioxidant and photoprotective activity. Finally, we suggested that MyNF might be a potential antioxidant ingredient in skincare products to prevent UVB-induced skin problems and also improve the cosmeceutical market values.

## Figures and Tables

**Figure 1 pharmaceutics-15-00906-f001:**
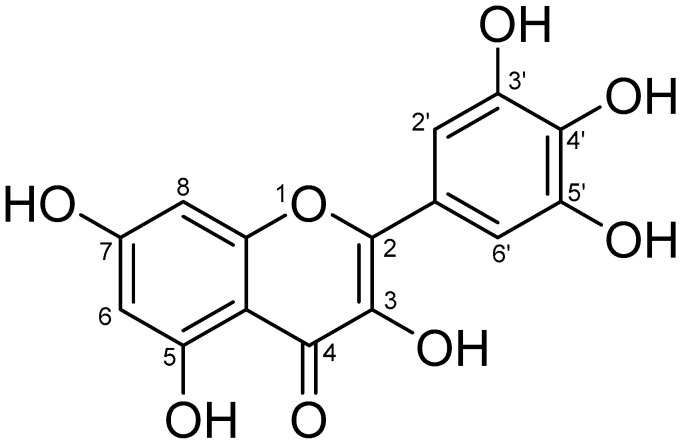
Chemical structure of myricetin (3, 3′, 4′, 5, 5′, 7-hexahydroxyflavone).

**Figure 2 pharmaceutics-15-00906-f002:**
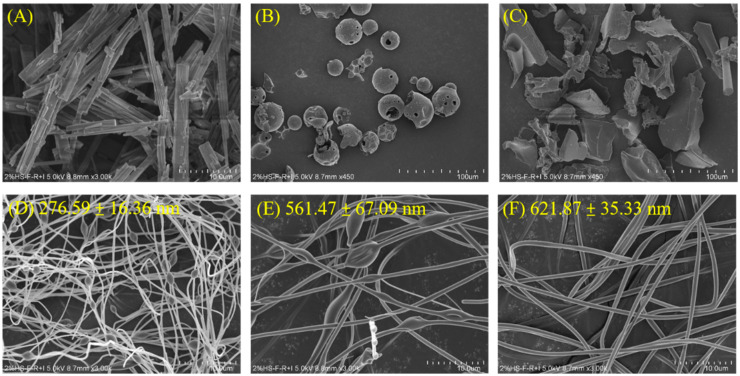
SEM images showing the surface morphology of (**A**) pure MYR, (**B**) pure HPBCD, (**C**) pure PVP, (**D**) MyNF 1:20:4, (**E**) MyNF 1:20:8, and (**F**) MyNF 1:20:12.

**Figure 3 pharmaceutics-15-00906-f003:**
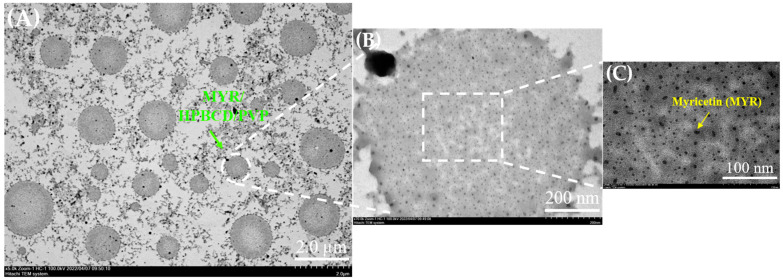
TEM images showing the inner morphology of myricetin nanofiber (MyNF 1:20:12) of (**A**) colloidal droplets of MYR/HPBCD/PVP, (**B**) PVP colloidal system containing dispersed MYR/HPBCD inclusion particles, and (**C**) a magnified view clearly revealing the conserved MYR/HPBCD size distribution.

**Figure 4 pharmaceutics-15-00906-f004:**
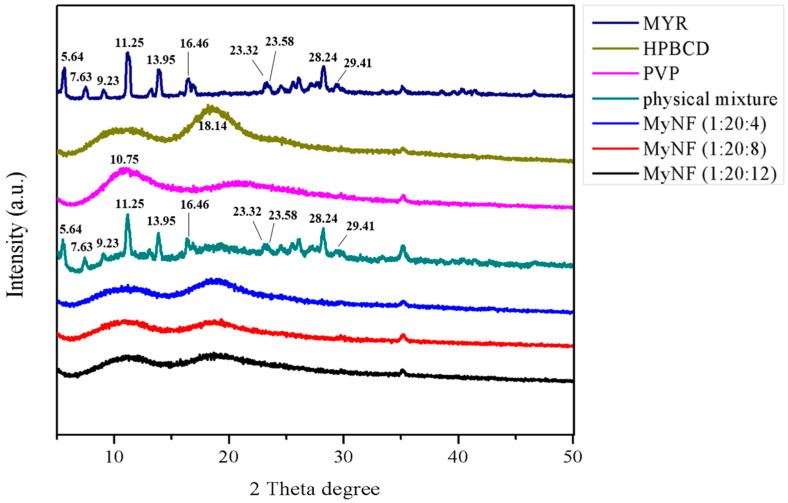
X-ray diffraction (XRD) spectra of MYR, HPBCD, PVP, physical mixture (MYR, PVP, and HPBCD powder), and the different ratios of MyNF (MYR:HPBCD:PVP; 1:20:4, 1:20:8, and 1:20:12; *w*/*w*/*w*).

**Figure 5 pharmaceutics-15-00906-f005:**
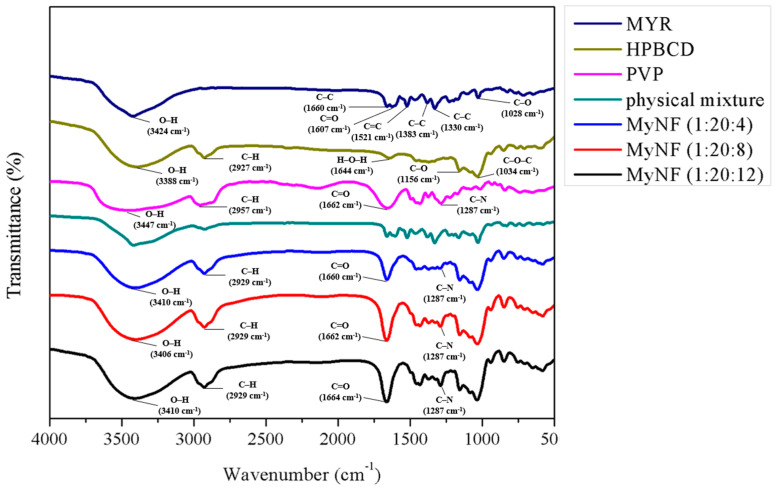
Fourier-transform infrared (FTIR) spectra of MYR, HPBCD, PVP, and physical mixture (MYR, PVP, and HPBCD powder), with different ratios of MyNF (MYR:HPBCD:PVP; 1:20:4, 1:20:8, and 1:20:12; *w*/*w*/*w*).

**Figure 6 pharmaceutics-15-00906-f006:**
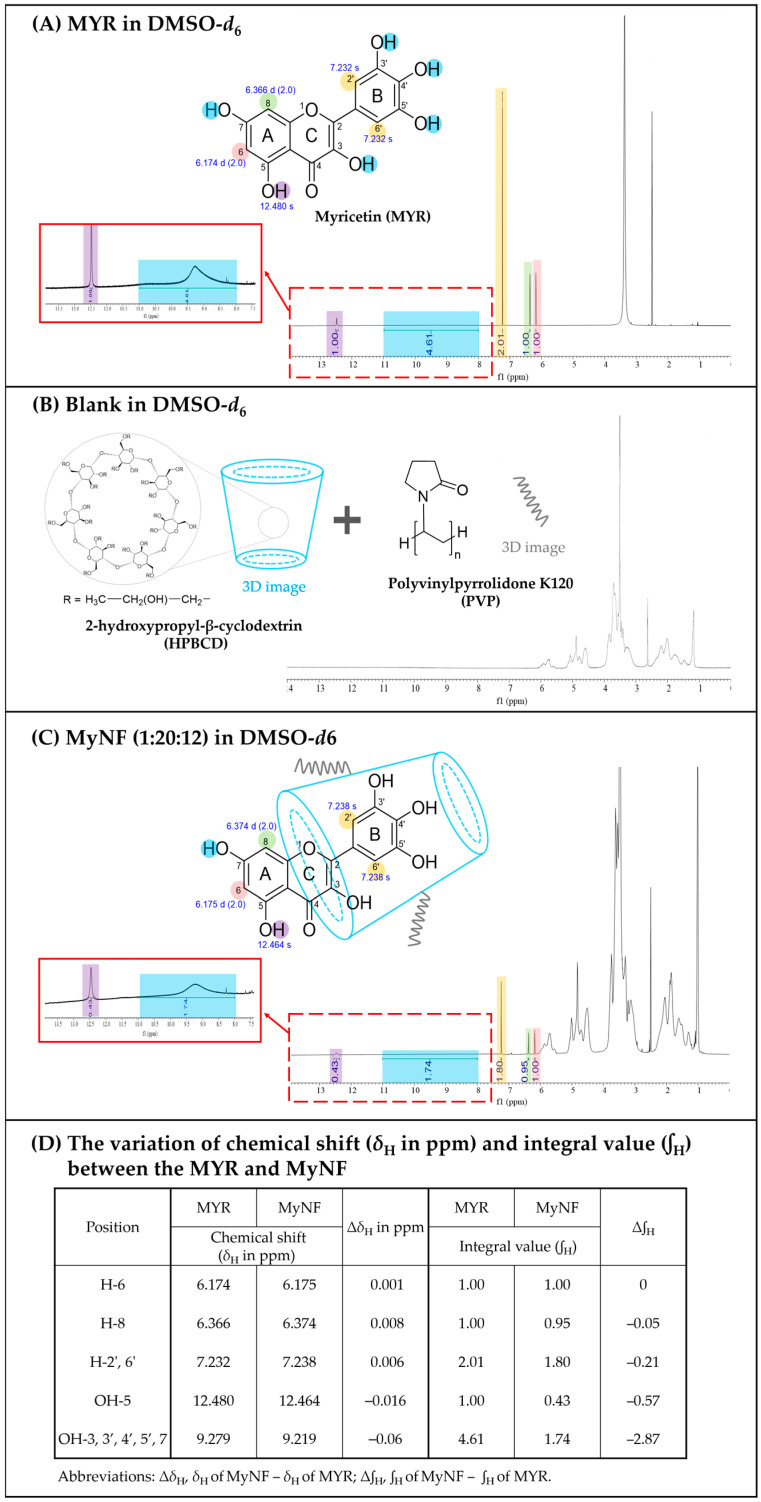
^1^H-nuclear magnetic resonance (^1^H-NMR) spectra of (**A**) MYR, (**B**) blank (HPBCD:PVP; 20:12; *w*/*w*), (**C**) MyNF (MYR:HPBCD:PVP; 1:20:12; *w*/*w*/*w*), and (**D**) variation of chemical shift (Δ*δ*_H_ in ppm) and integral value between the MYR and MyNF. Note: Pink color represents H-6; Green color represents H-8; Yellow color represents H-2′, 6′; Blue color represents OH-3, 3′, 4′, 5′, 7; purple color represents OH-5.

**Figure 7 pharmaceutics-15-00906-f007:**
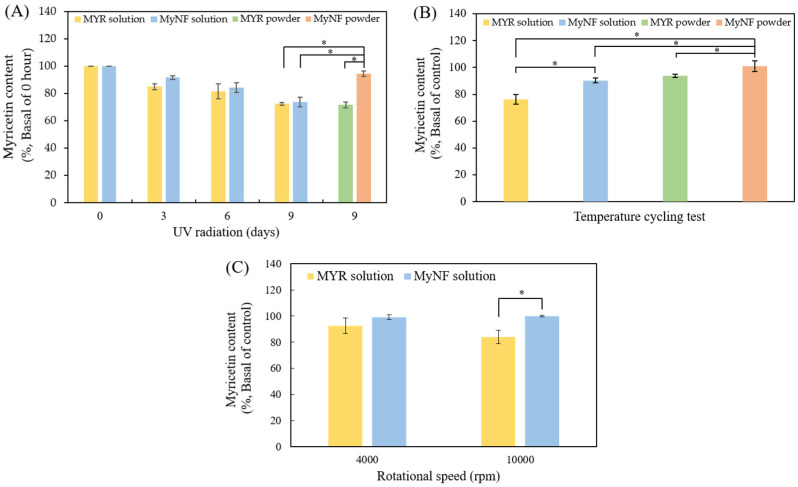
The stability test of photostability, thermostability, and centrifuge testing of MYR and MyNF (1:20:12). (**A**) The photostability of MYR solution, MyNF solution, MYR powder, and MyNF powder at 0 to 9 days of UV irradiation. (**B**) The thermostability of MYR solution, MyNF solution, MYR powder, and MyNF powder were tested at −20 °C, 4 °C, 25 °C, 50 °C, and 65 °C in sequence, each temperature for 24 h. (**C**) The centrifuge testing of MYR solution and MyNF solution, respectively, at 4000 and 10,000 rpm for 10 min. The values represent the mean ± SD (*n* = 3). * *p* < 0.05 was regarded as statistically significant.

**Figure 8 pharmaceutics-15-00906-f008:**
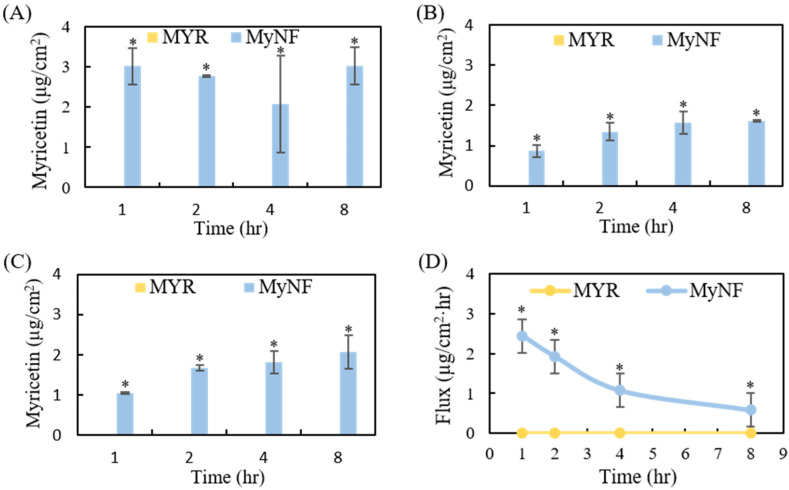
In vitro percutaneous penetration of MYR and MyNF: (**A**) stratum corneum, (**B**) epidermis, (**C**) dermis, and (**D**) flux of MYR and MyNF. The values represent the mean ± SD (*n* = 5). * *p* < 0.05 indicated a statistically significant difference compared with MYR.

**Figure 9 pharmaceutics-15-00906-f009:**
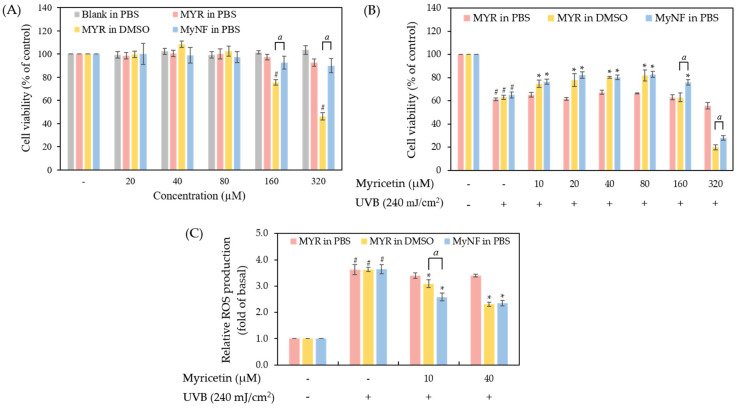
The cell viability and photoprotective effect of MYR and MyNF (1:20:12) in HaCaT keratinocytes. (**A**) The cell viability of HaCaT keratinocytes in pretreatment with MYR in PBS, MYR in DMSO, MyNF in PBS, and NF in PBS. (**B**) The photoprotective effect of MYR and MyNF against UVB-induced damage in HaCaT keratinocytes. (**C**) The ROS production in HaCaT cells in pretreatment with MYR and MyNF after UVB irradiation. All experiments were performed in triplicate and expressed as the mean ± SD (*n* = 3). “-” indicates the control group without any treatment and “+” indicates the UVB-irradiated group. # *p* < 0.05 compared with the control group. * *p* < 0.05 compared with the UVB-irradiated group. *^a^ p* < 0.05 was regarded as statistically significant between MYR in DMSO and MyNF in PBS.

**Table 1 pharmaceutics-15-00906-t001:** The water solubility, yield, and encapsulation efficiency of MyNF.

Ratio (MYR:HPβCD:PVP, *w*/*w*/*w*)	Water Solubility (µg/mL)	Yield (%)	Encapsulation Efficiency (%)
MYR	0.32 ± 0.16	-	-
1:20:4	852.83 ± 21.38 *^,#^	79.14 ± 2.89 *^,#^	99.54 ± 0.05 *
1:20:8	892.01 ± 10.11 *	86.30 ± 1.99 *	99.18 ± 0.02 *
1:20:12	914.67 ± 13.18 *	90.15 ± 5.04 *	98.63 ± 0.16 *

All experiments were performed in triplicate and expressed as the mean ± standard deviation (SD). * *p* < 0.05 indicates a statistically significant difference compared with MYR, and ^#^ *p* < 0.05 compared with MyNF (1:20:12).

**Table 2 pharmaceutics-15-00906-t002:** Particle size of MYR and its nanofibers detected by laser particle size analyzer.

Ratio (MYR:HPβCD:PVP, *w*/*w*/*w*)	Particle Size (nm)	Polydispersity Index (PDI)
MYR	4065.27 ± 327.11	1.37 ± 0.10
1:20:4	137.33 ± 6.79 *	0.10 ± 0.01 *
1:20:8	88.73 ± 3.07 *	0.07 ± 0.00 *
1:20:12	38.27 ± 1.02 *	0.24 ± 0.02 *

All experiments were performed in triplicate and expressed as the mean ± standard deviation (SD). * *p* < 0.05 indicates a statistically significant difference compared with MYR.

**Table 3 pharmaceutics-15-00906-t003:** The DPPH, ABTS^+^ free-radical scavenging, and reducing power of MYR in water, MYR in methanol, MyNF (1:20:12) in water, and vitamin C in water.

Group	DPPH Free Radical	ABTS^+^ Free Radical	FRAP
	SC_50_ (μg/mL) *^a^*		EC_50_ (μg/mL) *^b^*
MYR in water	NE *^c^*	NE *^c^*	NE *^c^*
MYR in methanol	10.70 ± 0.19 *^,#^	11.93 ± 0.66 *^,#^	3.79 ± 0.17 *^,#^
MyNF (1:20:12) in water	9.61 ± 0.10 *^,#^	11.31 ± 0.44 *^,#^	1.74 ± 0.64 *^,#^
Vitamin C in water *^d^*	14.35 ± 0.35 *^,#^	21.84 ± 0.80 *^,#^	5.95 ± 0.72 *^,#^

All experiments were performed in triplicate and expressed as the mean ± SD (*n* = 3). *^a^* SC_50_ was the concentration required to scavenge 50% of free radicals. *^b^* EC_50_ was concentration required effective reduction 50% of ferric irons. *^c^* NE indicated no effect. *^d^* Vitamin C was used as the positive control. * *p* < 0.05 indicated a statistically significant difference compared with MYR in water. ^#^ *p* < 0.05 compared with MyNF (1:20:12) in water.

## Data Availability

All data presented in the study are available on request from the corresponding author (flyen@kmu.edu.tw).

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
