# Peer review of "Myricetin Nanofibers Enhanced Water Solubility and Skin Penetration for Increasing Antioxidant and Photoprotective Activities"

_pharmaceutics, 2023, doi:10.3390/pharmaceutics15030906_

Round 1
Reviewer 1 Report
Dear Editor,
The manuscript entitled “Myricetin Nanofibers Enhanced Water Solubility and Skin Penetration for increasing Antioxidant and Photoprotective Activities” by Yen, F.-L. et al. describes the development of a myricetin nanofibers (MyNF) system of hydroxypropyl-ß-cyclodextrin (HPBCD)/polyvinylpyrrolidone K120 (PVP)-loaded with (myricetin) MYR for enhancement of water solubility and skin penetration by changing the physicochemical characteristics of myricetin (MYR). MYR is natural flavonoid compounds that significantly inhibited UV-induced keratinocyte damage but with limited bioavailability due to poor water solubility and inefficient skin penetration ability. MYR shows a variety of pharmacological activities such as antioxidant, anti-inflammatory, anti-photoaging, anti-allergic, and antimicrobial activities but its significant drawback is poor water solubility and poor water absorption. MyNF with different ratios of MYR:HPBCD:PVP (1:20:4, 1:20:8, and 1:20:12; w/w/w) 1were prepared. Water solubiliy, yield and encapsulation efficiency were determined by HPLC and particle size and morphology by SEM, DLS and TEM. XRD was used to determine the crystalline and amorphous structure of MYR and MyNF and determination of interaction studied by FTIR and 1H NMR while photostability was determined by exposure of MYR and MyNF to UV radiation and MYR monitoring by HPLC. The skin permeation of MYR through pork skin was determined using Franz diffusion cell (FDC). Antioxidant capacities were determined by DPPH and ABTS+ free radical scavenging activity as well as ferric reducing antioxidant power. HaCaT keratinoctyes were used for cell culture and cell safety assay. Effects of MYR and MyNF on photoprotective (UV) assay as well as UVB-Induced Reactive Oxygen Species (ROS) Assay was studied too.
MYR was practically insoluble in water and its solubility raised by up to 2858 times, by MYR:HPBCD:PVP formulation and encapsulation efficiency was high (above 98%). FTIR spectrum of physical mixture of MYR, HPCD and PVP did not show differences from component alone while FTIR of MyNF showed encapsulation of the aromatic ring into beta-CD and formation of hydrogen bonds attributed to interaction with PVP. Physical mixture also show distinctive XRD pattern of components while diffraction peaks of MYR disappeared in MyNF. Encapsulation increase photo and thermal stability of MYR particularly in dry form. MyNF exhibits excellent stability of centrifugation in solution skin penetration and photoprotection. In conclusion authors manage to prepare novel drug formulation (MyNF) showing increased solubility, stability, drug delivery over skin barriers. MyNF not only maintained the antioxidant activities of MYR, but also increased it due to improved water solubility.
In conclusion, in my opinion the manuscript is well structured and important and should be published in Pharmaceutics but after some minor corrections:
1 comment: In presented material 1H NMR of MyNF did not show any significant change in respect to MYR and blank. This fact should be stated clearly. Disappearance of phenolic OH at 13.49 ppm is dubious. Very small signal there is still visible although resolution is very low. Resolution of this picture/spectra (figure 6) should be increased too. In my opinion no change in 1H NMR is quite expected because 1H NMR is not very sensitive technic and dilution of MYR in respect to other components is very high. In addition, solution in d6-DMSO could restore components from MyNF by dissociation.
2 comment: I am not sure how deeply MyNF penetrates the skin intact. To my view probably just to some part and then MYR:HPBCD more profoundly. Can the authors comment about it? Also, what are advantages of MyNF in respect just to simpler encapsulation of MYR into HPBCD (MYR:HPBCD) alone? This issue was also not adressed in manuscript.
Minor comments:
Line 48 reference error
Line 242-243: Instead of phrase: “For the ABTS assay, the ABTS+ 2free radical solution was first mixed with 7 mM ABTS…” should be: “For the ABTS assay, the ABTS+ 2 free radical solution was first generated with 7 mM ABTS…”
Thank you for your collaboration.
Author Response
Thanks for the reviewer’s comment. The word file is our response to the questions raised by the reviewer.

Reviewer 2 Report
The manuscript entitled “Myricetin Nanofibers Enhanced Water Solubility and Skin Penetration for increasing Antioxidant and Photoprotective Activities”, authored Tzu-Ching Lin et al., deals with development of a nanofibers system of hydroxypropyl-β-cyclodextrin/polyvinylpyrrolidone loaded with Myricetin, and evaluation of its properties. The topic of this manuscript is important and current, and results could be interesting for readers. However, some changes have to be entered into the revised version of the manuscript before it can be further processed:
1. Figure 2 - please add a description of what the values in figures 2d-f refer to.
2. Some editorial errors occur in the text, e.g. line 48 - (Error! Reference source not found.). Please correct
Author Response

(The authors gave the same response as above.)

Reviewer 3 Report
1- Did the authors use any software for nanofibers dimension analysis such as image J?
2- The PDI values should range between 0.0 and 1.0 and this was stated by the authors in section 3.2.2. How did the authors report PDI values of 1.37 in Table 2? Pleas revise the readings.
3- Letters A, B, and C as well as the writing on Figure 3 are not clear.
4- Figure 6 is not obvious and is of low resolution. Please enlarge it on a separate page as a landscape not portrait so the details can be more clear.
Author Response

(The authors gave the same response as above.)
